# Dysregulated *TET* Family Genes and Aberrant 5mC Oxidation in Breast Cancer: Causes and Consequences

**DOI:** 10.3390/cancers13236039

**Published:** 2021-11-30

**Authors:** Bo Xu, Hao Wang, Li Tan

**Affiliations:** 1Center for Medical Research and Innovation, Shanghai Pudong Hospital, Fudan University Pudong Medical Center, and Shanghai Key Laboratory of Medical Epigenetics, Institutes of Biomedical Sciences, Fudan University, Shanghai 200032, China; 20211510009@fudan.edu.cn; 2Department of Anesthesiology, Zhongshan Hospital, Fudan University, Shanghai 200032, China

**Keywords:** breast cancer, *TET* genes, DNA methylation, 5mC oxidation, 5hmC

## Abstract

**Simple Summary:**

Both genetic and epigenetic mechanisms contribute to the pathogenesis of breast cancer. Since Tahiliani et al. identified TET1 as the first methyl-cytosine dioxygenase in 2009, accumulating evidence has shown that aberrant 5mC oxidation and dysregulated *TET* family genes are associated with diseases, including breast cancer. In this review we provide an overview on the diagnosis and prognosis values of aberrant 5mC oxidation in breast cancer and emphasize the causes and consequences of such epigenetic alterations.

**Abstract:**

DNA methylation (5-methylcytosine, 5mC) was once viewed as a stable epigenetic modification until Rao and colleagues identified Ten-eleven translocation 1 (TET1) as the first 5mC dioxygenase in 2009. *TET* family genes (including *TET1*, *TET2*, and *TET3*) encode proteins that can catalyze 5mC oxidation and consequently modulate DNA methylation, not only regulating embryonic development and cellular differentiation, but also playing critical roles in various physiological and pathophysiological processes. Soon after the discovery of TET family 5mC dioxygenases, aberrant 5mC oxidation and dysregulation of *TET* family genes have been reported in breast cancer as well as other malignancies. The impacts of aberrant 5mC oxidation and dysregulated *TET* family genes on the different aspects (so-called cancer hallmarks) of breast cancer have also been extensively investigated in the past decade. In this review, we summarize current understanding of the causes and consequences of aberrant 5mC oxidation in the pathogenesis of breast cancer. The challenges and future perspectives of this field are also discussed.

## 1. Introduction

Breast cancer is the most commonly diagnosed malignant tumor in women (2.3 million cases in 2020) and also one of the major causes of cancer-associated patient death (0.68 million deaths in 2020) worldwide [1]. The advances in the genetics and cell signaling studies have revolutionized our understanding of breast cancer pathogenesis and brought great progress in diagnosis and therapy. However, there is still a big gap in our understanding of the heterogeneity of breast cancer and their progressive metastasis and drug resistance, a process tightly associated with the plasticity of cancer genome in response to environmental changes. 

Epigenetic regulation, which links the extracellular environments to the genome, plays an essential role in cell fate determination and homeostasis maintenance. Accumulating evidence indicates that the epigenetic alterations including DNA methylation changes are involved in the pathogenesis of breast cancer [2]. DNA methylation has been once regarded as a stable epigenetic modification until Tahiliani et al. identified TET1 as the first methyl-cytosine dioxygenase [3]. TET proteins catalyze 5mC oxidation and generate cytosine modifications such as 5hmC/5fC/5caC, leading to active or passive DNA demethylation [3,4,5]. Although *TET1-MLL* translocation was observed in acute myeloid leukemia (AML) as early as 2003 [6], the exact biochemical activity of TET proteins and their functional roles have been extensively studied since 2009 (Figure 1). Here we present the current knowledge on the dysregulated *TET* family genes and aberrant 5mC oxidation in breast cancer as well as its contributions to many aspects of cancer hallmarks.

## 2. Aberrant 5mC Oxidation in Breast Cancer

5hmC/5fC/5caC are derived from 5mC, while their abundances are much less than 5mC in cells [5]. Interestingly, these 5mC derivatives (i.e., 5hmC) display much more dynamic changes in response to the fluctuation of physiological or pathophysiological conditions [7]. In 2011, Haffner et al. first reported that breast cancer tissues as well as other tumor tissues showed much lower 5hmC levels compared with their adjacent normal tissues [8]. Of note, the 5hmC level in primary cultured cells (including mammary epithelial cells) declines quickly during the in vitro culture [9,10], suggesting that the TET/5hmC epigenetic pathway is dramatically altered by the culture environment. Alternatively, the 5mC oxidation may be sensitive to the status of cell proliferation as Bachman et al. found that the 5hmC level is refractory to cell mitosis except for embryonic stem cells [11]. 

Aberrant 5mC oxidation (mainly indicated by the level of 5hmC) has potential diagnostic, prognostic, and predictive values for breast cancer patients. For instance, Tsai et al. found that 5hmC has prognostic value only in ERα-negative breast cancer [12]. In their study, lower global 5hmC levels in tumor tissues were associated with poorer prognosis of patients with ER/PR-negative subtype. In contrast, Wu et al. observed an increase of 5hmC in breast cancer [13]. The reports regarding the prognostic value of 5hmC in breast cancer are contradictory, suggesting that the prognostic value of 5hmC might be highly dependent on the subtypes of breast cancer. The abundance of 5fC and 5caC is much lower than that of 5hmC. There is only one study reporting the 5caC levels in breast cancer [14]. Unlike 5hmC, the levels of 5caC are elevated in some breast cancers (27%) and surprisingly, the intensity of 5caC does not correlate with that of 5hmC. The authors proposed that the unexpected 5caC increase in some breast cancers may be due to thymine DNA glycosylase (TDG) loss-of-function. 

In addition to the global changes of 5hmC/5fC/5caC levels, several labs have reported that the 5hmC levels on specific gene loci (i.e., *LZTS1* and *HLA-G*) declined in breast cancer [15,16]. By comparing the genome-wide 5hmC and 5mC distribution in basal-like breast cancers (BLBCs), Collignon et al. found that 5hmC gain in TET1-high breast cancers compared with TET1-low breast cancer was mostly associated with 5mC loss [17], suggesting that TET1 participates in the maintenance of DNA hypomethylation in these regions.

With the rapid development of liquid biopsy, the plasma cell-free DNA (cfDNA) in peripheral blood samples has become an alternative source for 5hmC detection. Given that a small percentage of cfDNA may be derived from tumor cells in cancer patients, it is rational to detect them according to cancer cell-specific epigenetic information (i.e., 5mC and 5hmC). Like 5mC, 5hmC also has the potential to be an epigenetic marker in the cfDNA assay for breast cancer patients. Although the prognostic value of 5mC in cfDNA has been widely studied, the role of cfDNA 5hmC in breast cancer is largely unknown [18]. According to the progress of cfDNA 5hmC detection in other malignancies (i.e., lung cancer, liver cancer, colorectal cancer, and pancreatic cancer) [18,19,20,21,22], it is promising to identify the breast cancer-specific 5hmC signatures in the plasma cfDNA of breast cancer patients.

Taken together, the reduction of the global 5hmC level is likely an epigenetic hallmark of breast cancer as compared to healthy mammary gland tissue; however, the diagnostic, prognostic, and predictive values of the global 5hmC/5caC/5fC levels in breast cancer remain elusive. It still requires further intensive research to identify the subtype- and stage-specific 5hmC signatures at specific gene loci in breast cancer. 

## 3. Dysregulation of *TET* Family Genes in Breast Cancer

### 3.1. Genetic Alterations of TET Family Genes in Breast Cancer

Among the three members of *TET* family genes (*TET1*, *TET2*, and *TET3*), *TET1* chromosome translocation and *TET2* mutations have been identified in hematopoietic malignancies such as AML and chronic myelomonocytic leukemia (CMML). Unlike the recurrent mutations in *PI3KCA*, *TP53*, *NOTCH*, and *BRCA1/2*, the genetic alterations of *TET* genes are rare in breast cancer. A recent GWAS analysis revealed a risk SNP associated with breast cancer is located within the *TET2* gene [23]. However, it is unclear whether this SNP influences the expression or function of the *TET2* gene.

Clonal hematopoiesis with indetermined potential (CHIP), a common age-associated genetic alteration, is associated with the increased risk of leukemia and cardiovascular diseases in elders [24,25]. Among the mutated genes, *TET2* is the No.2 gene with the highest mutation rates in elders [26]. However, it is unclear whether CHIP has any contribution to the development and progression of breast cancer in elder populations. Moreover, it is also unknown whether breast cancer cells have a similar mechanism of clonal expansion and genetic selection. Given that luminal breast cancer mainly occurs in old population, it is of great interest to investigate the relationship between the hematopoietic TET2 mutations and the initiation and progression of breast cancer.

### 3.2. Aberrant Expression of TET Family Genes in Breast Cancer

As early as 2011, Yang et al. reported that all three members of *TET* family genes were lowly expressed in multiple types of tumor tissues (including breast cancer) as compared with their adjacent normal tissues [27]. Similarly, Ruan and colleagues found that, of the 140 pairs of sample tissue, 95 (68%) exhibited lower levels of *TET1* mRNA in cancer tissues as compared with their normal-tissue control counterparts. Moreover, the *TET1* expression level was inversely correlated with breast cancer cell invasion and tumor development. In 2015, Shao and colleagues also confirmed that reduced expression of *TET* family genes, as well as TDG, are associated with poor prognosis of patients with early breast cancer [28]. However, there are also a few opposite reports. For instance, Wu et al. found that *TET1/3* expression levels were closely associated with tumor hypoxia, tumor malignancy, and poor prognosis in breast cancer patients [13]. In their study, they revealed that hypoxia conditions could upregulate *TET1/3* expression via HIF1α and in turn increase the 5hmC level in tumor cells. Indeed, they found *HREs* (Hypoxia response elements) in the upstream promoter regions of *TET1/3* genes. In 2019, Collignon et al. found that *TET1* is repressed by NF-κb in BLBC and high TET1 is associated with low levels of immune and defense response markers [17]. Estrogen signaling plays critical roles in luminal breast cancer and endocrine therapy has been successfully applied to the therapy of luminal breast cancer. Interestingly, several groups identified that estrogen signaling can upregulate *TET2* gene transcription through several distal enhancers [29,30].

Interestingly, epigenetic mechanisms also contribute to the aberrant expression of *TET* family genes during breast cancer pathogenesis. For instance, Sang et al. reported that *TET1* was downregulated and hypermethylated in highly metastatic breast cancer cell lines. Tao and colleagues found that EZH2 inhibitors could reactivate *TET1* expression in basal-like breast cancer cell lines, indicating that *TET1* gene might be epigenetically repressed by H3K27me3 in basal-like breast cancer [31]. Another issue of gene transcription that may be tightly associated with epigenetic mechanisms is the activation of alternative promoter. Good et al. identified an isoform of *TET1* driven by an alternative promoter in breast cancer, which yields a TET1 isoform without CXXC domain [32]. They found that the expression level of this TET1 isoform is high in triple-negative breast cancer (TNBC) and predicts poor prognosis. Similarly, an alternative promoter of the *TET2* gene was also reported in different tissues and cell lines including breast cancer cells although the biological significance of the new isoform remains unclear [33]. In addition, many miRNAs contribute to fine-tuning the expression of *TET* mRNAs. In 2013, Song et al. identified *TET* genes as targets of miR-22 in mammary tissues [34]. They found that miR-22 binds to the 3′UTR of *TET* mRNAs and represses their mRNA stability and expression. Later, dozens of miRNA have been identified as regulators of *TET* genes in different tissues/cells. However, little of them have been proved in breast cancer cells and a systematic evaluation is required to evaluate to what extent these miRNAs contribute to the dysregulated *TET* family genes in breast cancers.

The levels of TET proteins are also precisely controlled at the post-translational layer. For instance, acetylation of TET2 as well as other TET members may be regulated by p300 and HDAC1/2 [35]. Phosphorylation of TET2 at S99, which is regulated by AMPK and PP2A, stabilize the TET2 protein [36,37,38]. Phosphorylation of TET2 at T1939 and T1964, which are regulated by the JAK2 pathway, influence the inflammatory signaling [39]. Autophagy is a strategy to control the recycling of macromolecules. As aforementioned, p53 could modulate the TET2 protein level through autophagy [40]. However, it requires further study to determine to what extent these post-translational modifications and corresponding signaling pathways contribute to the deregulated expression of *TET* family genes in breast cancer.

### 3.3. The Rewired Catalytic Activity of TET Proteins in Breast Cancer

Several co-factors, such as α-KG, oxygen, iron, and ascorbic acid, are required for TET proteins-catalyzed 5mC oxidation [7]. The changes in the intracellular concentrations of these co-factors influence the enzymatic activity of TET proteins. The IDH1/2 mutants, encoded by two recurrent mutated genes in glioma and AML, generate 2HD, so-called “oncometabolite”, which competes for α-KG and inhibits TET catalytic activity [41]. SDH and LDH mutants that influence the metabolism of succinate also inhibit TET activity [42]. In contrast, transient increase in α-KG concentration by glucose or glutamine shock increases 5hmC in vitro and in vivo [43]. Given that high glucose intake and consumption are features of most cancer cells, it will be of great interest to investigate whether the “Warburg effect” has any direct effect on 5mC oxidation catalyzed by TET proteins. 

Ascorbic acid (also known as vitamin C) could recycle the iron for TET proteins-catalyzed 5mC oxidation [44]. Ascorbic acid has been applied to treat different cancer cells in a dish or mice and achieved exciting success. For instance, ascorbic acid treatment could reverse the leukemogenesis in mice by restoring the function of TET proteins [45]. A recent study has reported that ascorbic acid treatment could increase the 5hmC level and activate apoptosis gene expression such as *TRAIL* in breast cancer cells [46]. In theory, ascorbic acid treatment could increase 5hmC level and facilitate the maintenance of a healthy DNA methylome partially through modulating TET-catalyzed 5mC oxidation and DNA demethylation. Since ascorbic acid can be easily supplied as a nutrient in food (i.e., vegetables and fruits), more evidence is required to determine whether and how ascorbic acid could provide benefits in the prevention and therapy of breast cancer and to what extent such benefits are dependent on TET proteins-mediated 5mC oxidation.

Hypoxia is a common feature of solid tumor microenvironment due to the rapid growth of tumor cells and insufficient vascular blood supply. Since oxygen is essential for the oxidative reaction, low concentration of oxygen in tumor microenvironment could inhibit the catalytic activity of TET proteins, leading to a low 5hmC level in tumor cells [47]. This finding is completely opposite to the result of Wu et al. that hypoxia upregulates TET1/3 expression via HIF1α and thereby increases the 5hmC level [13], suggesting that TET has a complicated relationship with hypoxia. On one hand, *TET1* gene is a target of HIF1α. There are a few *HREs* in the upstream regulatory region of *TET1* gene and acute hypoxia can induce *TET1* transcription. On the other hand, TET1 protein could synergize HIF1α to activate the expression of HIF1α target genes. However, hypoxia (if the O_2_ concentration is lower than the Km) itself may impair TET proteins-catalyzed 5mC oxidation. Given that sometimes these mechanisms may reach the opposite effect, it will be of importance to distinguish which mechanism plays the dominant role in certain conditions.

### 3.4. Aberrant Genomic Targeting of TET Proteins in Breast Cancer

Both TET1 and TET3 contain CXXC domain in their N-termini, which can facilitate their binding to CpG islands and thereby protect the CpG islands from DNA hypermethylation [48,49,50]. Although TET2 does not contain CXXC domain, CXXC4/IDAX and CXXC5/RINF, encoded by *CXXC4* (a neighbor gene of *TET2*) and *CXXC5,* respectively, can bind to TET2 and facilitate its chromatin recruitment to CpG islands but also trigger the cleavage of TET2 by Caspase 3 [51]. In addition to the CXXC domain which prefers binding to DNA sequence with high-density GpG, transcriptional factors could interact with TET proteins and recruit them to specific gene loci for 5mC oxidation and DNA demethylation. In different tissues, various TFs have been identified to recruit TET2 protein to their unique genomic targets [52,53]. In breast cancer, FOXA1 which activates *TET1* gene expression could also recruit TET1 to a subgroup of enhancer regions [54]. Therefore, the aberrant expression of such TFs or their mutations may change the action mode of TET proteins-mediated 5mC oxidation in breast cancer.

As a female prevalent cancer, breast cancer has tight connection with estrogen signaling. Activation of the nuclear receptor ERα and subsequent ERα-controlled transcriptome is the primary molecular mechanism through which estrogen exerts its physiological and pathophysiological functions. In fact, around 70% breast cancers belong to ERα+ breast cancer (so-called luminal A or B subtypes). It has been revealed that ERα cistrome and transcriptome were reprogrammed during the pathogenesis and progression of ERα+ breast cancer cells. A recent report found that TET2 functions as a co-activator of ERα and regulates the activity of enhancers containing estrogen response elements (*EREs*) through DNA demethylation [30]. Broome et al. proved the interaction between TET2 and ERα/GATA3 complex in breast cancer cells [55]. Co-localization of TET2 and ERα contributed to the maintenance of 5hmC at ERα-targeted enhancers, which regulated expression of ERα-targeted genes. Meanwhile, disrupting the TET2/ERα/GATA3 complex resulted in the dysregulation of cell cycle related gene expression. Our recent work also showed that TET2 loss led to DNA hypermethylation on a large proportion of enhancers including *EREs* in MCF7 cells [56]. Consistent to the notion that CpG methylation within *EREs* impairs ERα binding, our ERα ChIP-seq demonstrated that loss of TET2 impaired the ERα binding and gene transcription of a subgroup of E2-responsive genes in MCF7 cells. However, unlike several previous studies, we did not detect the binding between ERα and TET2 by co-IP experiment, indicating that other molecule (s) might be responsible for the recruitment of TET2 to these enhancers. Collectively, TET2-mediated 5mC oxidation is involved in the regulation of DNA methylation and transcriptional activity of certain enhancers including *EREs*. However, the detailed molecular mechanisms through which TET2 is recruited to these enhancers and how these different events are coordinated remain controversial and require further study.

## 4. Consequences of Dysregulated *TET* Family Genes and Aberrant 5mC Oxidation in Breast Cancer

Given the critical role of TET proteins-catalyzed 5mC oxidation in the regulation of DNA methylation dynamics, the dysregulation of *TET* family genes may participate in the development and progression of breast cancer. At the molecular level, aberrant 5mC oxidation could partially contribute to the well-known epigenetic hallmark of cancer (i.e., global DNA hypo-methylation but local DNA hypermethylation) [7]. In addition, TET proteins could also exert catalytic activity-independent function in the regulation of chromatin structure and gene transcription [7]. Here, we summarize the broad impact of dysregulated *TET* family genes and aberrant 5mC oxidation on multiple aspects (cancer hallmarks) of breast cancer (Figure 2).

### 4.1. Genomic Instability and Mutation

Chromosome abnormality and gene mutations are accumulated during the initiation and progression of breast cancer. DNA methylation plays essential role in the maintenance of genomic stability through epigenetic silencing of repeated elements [57]. In addition, the TDG-mediated base excision repair (BER) pathway as well as other DNA repair pathways (DNA deamination and glycosylation) have been proposed to work together with TET/5mC oxidation [4,58]. Cellular immunostaining and hMeDIP-seq analysis revealed that 5hmC signal was enriched in the regulatory and transcribed regions within the euchromatin regions but depleted from the heterochromatin regions [49,59,60]. The differential patterns of genomic 5hmC and 5mC distribution suggest that only the 5mC within the relatively open chromatin regions could undergo TET proteins-catalyzed oxidative reaction while the 5mC within the heterochromatin regions might be protected by certain mechanisms. Unexpectedly, there are several reports showing that the repeated regions (i.e., LINE1) might be activated by TET1/2 proteins [61,62,63]. However, a recent report showed that TET2 depletion leads to DNA hypomethylation in the intergenic repeated regions [64]. The authors speculate that this unexpected phenomenon might be caused by the redistribution of Dnmt3a/b from heterochromatin to euchromatin due to the loss of Tet2 competition in euchromatin.

### 4.2. Stemness

Loss of TETs results in DNA hyper-methylation and epigenetic silencing of *miR-200* which leads to epithelial-mesenchymal transition (EMT) and increases the stemness of breast cancer cells [34]. Moreover, TET2 may suppress breast cancer stem cells (CSC) through the regulation of the miRNA200c/PKCζ axis [65]. The comparison of wild-type and *Tet2^−/−^* mice also revealed that loss of Tet2 enhances the self-renewal of mammary stem cells (MaSCs) while impairing the luminal lineage commitment [66]. However, there were also some opposite reports. Wu et al. revealed that increased TET1/3 levels in breast cancers are associated with TNF-α expression, which activates the p38-MAPK pathway and maintains breast tumor-initiating cells (BTIC) [13]. Slow-cycling cancer cells (SCCCs) were believed as the reservoir of chemoresistance during cancer progression. Puig and colleagues identified TET2 as a marker for the stem-like cell in tumors and showed that 5hmC was enriched in the persisting SCCCs [67]. Therefore, it remains a debate on whether *TET* family genes play a positive (oncogenic) or negative (tumor suppressive) role in the maintenance of cancer stem cells 

### 4.3. Cell Proliferation

Hormone positive breast cancer cells showed addiction to E2/ER signaling. Ali and colleagues found that loss of TET2 impairs E2/ERα-stimulated cell growth [30], indicating an oncogenic role of TET2 in ERα+ breast cancer. In their study, TET2 acts as a co-activator to promote E2/ERα-induced gene transcription via DNA demethylation and MLL-dependent H3K4me1/2 generation at enhancers. Moreover, E2/ER promotes *TET2* gene transcription in MCF7 cells due to the binding of ERα to 3 ERE upstream of the *TET2* gene, suggesting a positive feedback between ERα mediated gene transcriptional activation and TET2-mediated enhancer DNA demethylation. Conversely, our work suggests a tumor-suppressive role of TET2 in ERα+ breast cancer [56]. In our work, TET2 depletion has no significant effect on the adherent cell growth while promotes the anchorage-independent growth of MCF7 cells. Further epigenomic and transcriptomic analyses revealed that TET2 depletion led to selective enhancer DNA methylation and subsequent impairment of TF binding and gene transcription. Interestingly, some apoptotic death genes were among these enhancers/genes, indicating that TET2 functions as a brake or negative feedback in E2/ERα signaling by maintaining the expression of those negative regulators.

### 4.4. Invasion and Metastasis

The first report on the functional role of *TET* genes in breast cancer was from Ruan and colleagues [68]. In 2012, they demonstrated a tumor-suppressive role of TET1 in breast cancer as knockdown of *TET1* by shRNA promotes the invasion and metastasis of breast cancer cells. Mechanistically, they found that TET1 could promote the DNA demethylation on the promoters of *TIMP2/3* genes, which encode the inhibitors of MMPs and inhibit extracellular matrix degradation [68]. Soon after, Sun et al. reported that TET1 activates *HOXA9* gene via DNA demethylation, and the latter is involved in the regulation of migration and metastasis [69]. Pandolfi and colleagues found that TET proteins activate the expression of *miRNA-200* via DNA demethylation of *mir-200* promoter [34]. Low expression of TET genes leads to decreased miRNA-200a/b/c and enhanced epithelial–mesenchymal transition (EMT) of breast cancer cells, an important process not only in stemness but also in invasion and metastasis. TET2 could inhibit the migration and invasion of breast cancer cells through the demethylation of *EpCAM* and *E-cadherin,* again preceding their expression and activation [70].

### 4.5. Cell Death

The functional roles of *TET* family genes in cell death induced by chemotherapy and radiotherapy are controversial. A recent study showed that TET2 interacts with promyelocytic leukemia (PML), which plays a critical role in cell-cycle arrest, senescence, apoptosis, genome stability, and antiviral effects [71]. Loss of TET2 is associated with increased resistance of cancer cells to chemotherapy in vitro and in vivo [71]. Moreover, Sant et al. found that vitamin C, which activates TET family proteins, could induce apoptosis in several breast cancer cell lines through upregulating the expression of *TRAIL* [46,72]. However, another study in the neck and head tumor revealed that high 5hmC and increased TET2 expression are associated with dormancy and drug resistance to chemotherapy while knockdown of TET2 eliminates the dormant cancer cells [67]. As TET2 was downregulated by p53-mediated autophagy, p53 loss-induced TET2 increase facilitates the acquisition of chemoresistance in cancer cells [40]. Therefore, the molecular mechanism underlying this discrepancy is not fully understood and requires further in-depth research.

### 4.6. Inflammation and Tumor Immunity

The role of epigenetic regulation in inflammation and tumor immunity has attracted a lot of attention. Collignon et al. observed an anti-correlation between *TET1* expression level and the extent of infiltration by the major types of leukocytes in BLBC, while such anti-correlation did not exist in other subtypes of breast cancers [17]. Together with the findings that TET1 expression is increased in some BLBC, these data indicate that TET1 may play an oncogenic role in BLBC and such oncogenic role is partially associated with decreased inflammatory response and anti-tumor response. Again, their findings also reinforce a subtype-specific role of TET1 in modulating the inflammation and cancer immunity in breast cancer.

Xiong and colleagues found that TET2 is essential for IFN-gamma induced chemokine gene expression in melanoma and colon cancer cells [73]. As one of the IFN-gamma induced genes, *PD-L1* was also enhanced by TET2. Conversely, our recent work revealed that TET2 represses the expression of the *PD-L1* gene in breast cancer either in the presence or absence of IFN-gamma treatment [74]. Mechanistically, we found that TET2-mediated *PD-L1* repression was independent of DNA methylation, instead of the recruitment of HDAC1/2 by TET2 to the promoter of *PD-L1* gene. The inhibitors of DNMTs, such as 5-Aza, could activate endogenous virus expression, which in turn activate type-I IFN response, converting the “cold” tumor to “hot” tumor. Given that TET2 knockout leads to intergenic DNA hypomethylation, it is possible that TET proteins also modulate the responsiveness of tumor cells to immunotherapy through similar mechanisms.

In addition to the well-studied cell-autonomous role of TET2 in tumor cells, our lab also revealed that the comprehensive knockout of *Tet2* in mice leads to an impaired anti-tumor immunity due to the overactivation of granulocytic myeloid-derived suppressor cells (G-MDSCs) which subsequently decrease the number of cytotoxic CD8^+^ T cells [75]. Therefore, TET family genes regulate anti-tumor immunity of breast cancer from not only the aspect of cancer cells but also the non-tumor cells in tumor microenvironment especially the immune cells. Since targeting the immune checkpoint achieved great success in clinical cancer therapy including the TNBCs, it will be of great interest to investigate whether TETs are new targets to improve the anti-tumor immunity and immunotherapy efficiency of breast cancer.

## 5. Conclusions

As the initial step of DNA demethylation, TET proteins-catalyzed 5mC oxidation regulates the stability and plasticity of the epigenomes of mammary tissues. The dysregulation of *TET* family genes and aberrant 5mC oxidation are involved in breast cancer pathogenesis. Of note, the value of 5hmC/5fC/5caC and *TET* gene expression in the diagnosis and prognosis of breast cancer is largely dependent on the subtypes of breast cancer, suggesting a double-edged sword role of TET/5mC oxidation pathway in breast cancer. Therefore, caution should be taken when distinguishing the exact role of *TET* family genes in different subtypes, aspects, and stages of breast cancer before we transform this knowledge in the diagnosis and therapy of breast cancer.

## Figures and Tables

**Figure 1 cancers-13-06039-f001:**
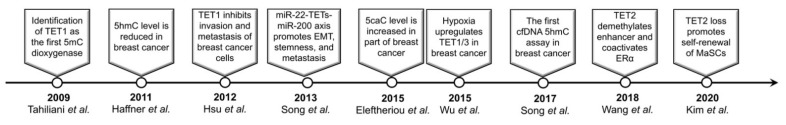
Milestones of the research for TET/5mC oxidation in breast cancer.

**Figure 2 cancers-13-06039-f002:**
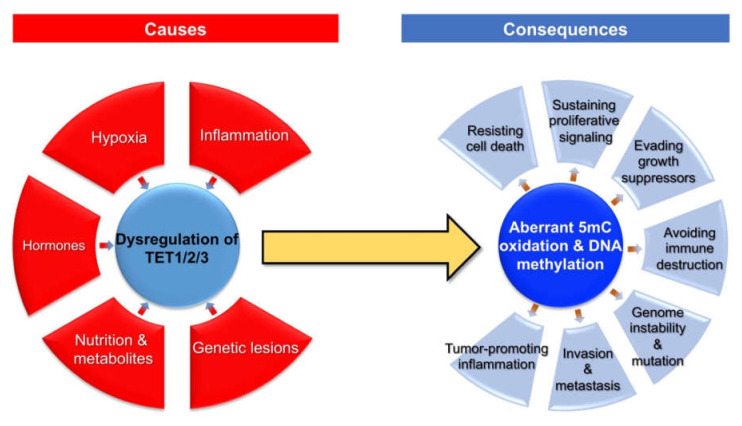
Causes and consequences of aberrant 5mC oxidation in breast cancer.

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
