# Peer review of "Dysregulated TET Family Genes and Aberrant 5mC Oxidation in Breast Cancer: Causes and Consequences"

_cancers, 2021, doi:10.3390/cancers13236039_

Round 1

Reviewer 1 Report

This is a well-written review that provides a comprehensive overview of the state of knowledge regarding the mechanisms controlling DNA methylation through the level of 5methyl Cytosine (5mC) and their alteration in the context of breast cancer. The authors notably present the current understanding of the causes and consequences of aberrant 5mC oxidation in the pathogenesis of breast cancer.

I have no particular comment except in the section dealing with cell death where, curiously, the authors exclusively cite examples outside the context of breast cancer. They might for instance consider discussing the results of Sant et al, regarding Trail and apoptosis, which they actually quote in paragraph 3.3.

Author Response

Thanks a lot for your insightful comments and professional suggestion. According to your suggestion, we have added the work of Sant et al. in the "4.5 Cell death” part (line 371-373, Page 8).

Reviewer 2 Report

In this review, the authors summarize current knowledge of aberrant 5 mC oxidation and dysregulation of TET genes in breast cancer and discuss their potential diagnosis and prognosis values for breast cancer development and treatment. With this regard, the inclusion of “TET” in the title might be more appropriate.  The review is well written and organized. Although no clear conclusion can be draw for the important topic at this moment, it is necessary to publish this review for attracting more research in the field.

Author Response

Thank you so much for your valuable suggestions. According to your advice, we have changed the title into “Dysregulated TET family genes and aberrant 5mC oxidation in breast cancer: causes and consequences”.